# Irritable Bowel Syndrome with Diarrhea (IBS-D): Effects of *Clostridium butyricum* CBM588 Probiotic on Gastrointestinal Symptoms, Quality of Life, and Gut Microbiota in a Prospective Real-Life Interventional Study

**DOI:** 10.3390/microorganisms13051139

**Published:** 2025-05-15

**Authors:** Francesco Di Pierro, Fabrizio Ficuccilli, Laura Tessieri, Francesca Menasci, Chiara Pasquale, Amjad Khan, Fazle Rabbani, Nazia Mumtaz Memon, Massimiliano Cazzaniga, Alexander Bertuccioli, Mariarosaria Matera, Ilaria Cavecchia, Martino Recchia, Chiara Maria Palazzi, Maria Laura Tanda, Nicola Zerbinati

**Affiliations:** 1Microbiota International Clinical Society, 10123 Turin, Italy; f.dipierro@vellejaresearch.com (F.D.P.); pchiaramaria@gmail.com (C.M.P.); 2Scientific & Research Department, Velleja Research, 20125 Milan, Italy; maxcazzaniga66@gmail.com; 3Department of Medicine and Technological Innovation, University of Insubria, 21100 Varese, Italy; nicola.zerbinati@uninsubria.it; 4Nursing Home Quisisana, 00197 Rome, Italy; f.ficuccilli@gmail.com (F.F.); lauratessieri@yahoo.it (L.T.); francescamenasci@gmail.com (F.M.); chiara.pasquale@virgilio.it (C.P.); 5Department of Biochemistry, Liaquat University of Medical & Health Sciences (LUMHS), Jamshoro 76090, Pakistan; 6Department of Psychiatry, Lady Reading Hospital (LRH), Peshawar 25000, Pakistan; 7Department of Pathology, Liaquat University of Medical and Health Sciences (LUMHS), Jamshoro 76090, Pakistan; naziamumtaz@lumhs.edu.pk; 8Department of Biomolecular Sciences, University of Urbino Carlo Bo, 61029 Urbino, Italy; alexander.bertuccioli@uniurb.it; 9Department of Pediatric Emergencies, Misericordia Hospital, 58100 Grosseto, Italy; jajamatera74@gmail.com; 10Microbiomic Department, Koelliker Hospital, 10134 Turin, Italy; 11Unit of Clinical Epidemiology and Biostatistics, Mario Negri Institute Alumni Association (MNIAA), 20156 Milan, Italy; statmed@hotmail.com; 12Endocrine Unit, Department of Medicine and Surgery, University of Insubria, 21100 Varese, Italy; marialaura.tanda@uninsubria.it

**Keywords:** *Clostridium butyricum* CBM588, irritable bowel syndrome, IBS-D, butyrate producers

## Abstract

Diarrhea-predominant irritable bowel syndrome (IBS-D) is a functional gastrointestinal disorder characterized by altered motility, abdominal pain, and dysbiosis—particularly reduced biodiversity and a lower abundance of butyrate-producing bacteria. Strategies that modulate the gut microbiota may offer therapeutic benefit. *Clostridium butyricum* (*C. butyricum*) CBM588 is a butyrate-producing probiotic with immunomodulatory properties and potential efficacy in treating gastrointestinal disorders. This pragmatic, prospective, open-label, single-arm interventional study assessed the clinical, microbial, and safety-related effects of an 8-week CBM588 supplementation, along with a low-fiber and low-residue diet, in 205 patients with IBS-D who attended Quisisana Nursing Home Hospital, Rome, Italy, between November 2024 and February 2025. The primary outcomes included the global symptom response, the Bristol Stool Scale (BSS), stool frequency, diarrhea episodes, abdominal pain (severity and frequency), bloating, bowel dissatisfaction, quality of life (QoL), safety, and treatment tolerability—measured using the IBS Symptom Severity Scale (IBS-SSS) and a standardized tolerability scale. CBM588, in patients treated with a low-fiber and low-residue diet, significantly improved all clinical endpoints, with a >80% reduction in diarrhea episodes; ~60% reductions in stool frequency and abdominal pain; and >50% improvements in bloating, bowel dissatisfaction, and QoL. Treatment was well tolerated (mean tolerability score 8.95 ± 0.88), with >95% adherence, and no serious adverse events were reported. The secondary outcomes included changes in gut microbiota. In a subset of patients, 16S rRNA gene sequencing showed increased α-diversity and enrichment of butyrate-producing genera (*Agathobacter*, *Butyricicoccus*, *Coprococcus*), which correlated with symptom improvement. Bloating increased in some patients, possibly related to fermentation activity. These findings support the *C. butyricum* CBM588 probiotic strain as a safe, well-tolerated, and microbiota-targeted intervention for IBS-D. Randomized controlled trials are warranted to confirm efficacy.

## 1. Introduction

Irritable bowel syndrome (IBS) is one of the most common gastrointestinal (GI) problems affecting of approximately 4.4–4.8% of the population in industrialized nations [1,2]. IBS is characterized by abdominal pain and defecation-associated disorders, with changes in bowel habits affecting patients’ social lives and quality of life (QoL) [3]. IBS is classified into four subtypes: constipation-predominant (IBS-C), diarrhea (IBS-D), mixed (IBS-M), and undefined (IBS-U) [4]. The classification of and therapy for IBS are commonly established according to the Bristol Stool Form Scale (BSFS) and Rome IV criteria [5,6]. Although the etiology and pathophysiology of IBS have not been clarified [7], a putative contributing factor seems to be alterations in the gut microbiota [8]. Particularly, in IBS-D, the gut microbiota appears to be characterized by a reduction in both butyrate producers and α-biodiversity [9]. Butyrate producers play many physiological roles inside the gut microbiota in favor of the host. Locally produced butyrate feeds colonocytes, maintains gut barrier integrity, limits immune pro-inflammatory gut cascades, and inhibits oncogenic pathways [10]. Intriguingly, a reduced microbiota diversity deprives the gut microbiota of its capacity to metabolize acetate to butyrate, mainly affecting the acetyl-CoA pathway [11]. Moreover, a reduction in colonic butyrate leads to lower sodium and water reabsorption capacities, favoring the development of diarrhea [12].

*Clostridium butyricum* (C. *butyricum*) is a butyrate-producing human gut symbiont that has been safely used as a probiotic for decades. It has been investigated in gut-acquired infections, intestinal injury, IBS, inflammatory bowel disease, neurodegenerative diseases, metabolic disorders, and colorectal cancer [13]. A *C. butyricum*-based probiotic administered to patients with IBS-D was clinically shown to significantly counteract overall symptoms, QoL, and stool frequency in Chinese patients [14]. *C. butyricum* MIYAIRI 588 (also referred to as *C. butyricum* FERM BP-2789), a probiotic strain studied for over 50 years, mainly in Asia [15], was authorized by the European Parliament in 2014 a as Novel Food [16].

The present study pragmatically evaluated the effects of *Clostridium butyricum* CBM588, the only butyrate-producing probiotic currently available for medical use in Europe, in an Italian cohort of patients diagnosed with IBS-D. Additionally, a subset of participants underwent gut microbiota analysis to examine their microbial composition changes and responses to the probiotic treatment.

## 2. Materials and Methods

### 2.1. Study Design

This was a prospective, single-center, single-arm, open-label, pragmatic interventional clinical study designed to evaluate the perceived efficacy and safety of the *Clostridium butyricum* CBM588 probiotic strain in patients with diarrhea-predominant irritable bowel syndrome (IBS-D), conducted at Quisisana Nursing Home Hospital, Ferrara, Italy. Participants were enrolled in a real-life clinical setting, reflecting routine care conditions. Due to its integration into routine practice; broad inclusion criteria; and open-label, non-comparative format, this study followed a pragmatic trial design aimed at assessing effectiveness under everyday clinical conditions rather than establishing causal inference. Given the single-arm interventional design of this study, clinical outcomes were contextualized through a comparison with a retrospective cohort of patients with IBS-D who had previously received standard care at the same hospital.

This study was approved by the Ethics Committee for Human Experimentation, University of Urbino Carlo Bo, Urbino, Italy (approval date: 23 May 2024), and conducted in accordance with the Declaration of Helsinki. All participants provided informed written consent before enrolling in this study. This study was registered with the clinicaltrial.gov registry (Reg. number NCT06676514).

### 2.2. Primary Outcomes

The primary outcomes of this study included IBS symptom severity, stool consistency and frequency, quality of life (QoL), and treatment safety and tolerability. These were assessed at baseline and after 8 weeks of supplementation using the IBS—Symptom Severity Scale (IBS-SSS), the Bristol Stool Score (BSS), the IBS Quality of Life (IBS-QoL) questionnaires, and an internally standardized tolerability scale.

The IBS-SSS, with a total score range of 0–500, evaluates five domains: abdominal pain severity and frequency, bloating, satisfaction with bowel habits, and the overall impact of symptoms on QoL. IBS severity was categorized as mild (<175 points), moderate (175–300 points), or severe (>300 points). A reduction of ≥50 points from baseline was considered a clinically meaningful treatment response [17]. The BSS was assessed to evaluate stool shape and stool consistency. It categorizes stool into seven types, ranging from hard lumps (Type 1), indicating constipation, to watery stool (Type 7), associated with diarrhea [18]. Quality of life was assessed using the IBS-QoL, a validated 34-item questionnaire scored from 0 to 100, with higher scores indicating a better QoL. The scale includes eight dimensions: dysphoria, activity interference, body image, health concerns, food avoidance, social reactions, sexual concerns, and interpersonal relationships [19].

Safety and tolerability were evaluated using a 10-point standardized scale (maximum tolerability, 10; good tolerability, 7–9; moderate tolerability: 4–6; poor tolerability, 1–3; no tolerability, 0) and continuous clinical monitoring. To support patient safety and address concerns in real time, each participant was provided with direct 24/7 access to a study physician. Continuity of care was maintained throughout this study, including for patients who discontinued participation. Adherence to the probiotic regimen was recorded as a percentage based on patient-maintained daily diaries.

### 2.3. Secondary Outcomes

The secondary outcome in this study was the analysis of gut microbiota modifications associated with *C. butyricum* CBM588 supplementation. Microbiota analysis was conducted on a subset of 19 randomly selected participants who provided stool samples at baseline and after the 8-week treatment period. We used 16S rRNA gene sequencing to assess changes in microbial composition and α-diversity, with particular attention to shifts in butyrate-producing bacterial taxa and their potential associations with clinical improvements.

### 2.4. Participants and Criteria

This study was conducted in a real-life clinical setting as a pragmatic trial without a prospective control group. Outpatients of both sexes, aged ≥18 years, with a diagnosis of diarrhea-predominant irritable bowel syndrome (IBS-D) were enrolled at Quisisana Nursing Home Hospital, Ferrara, Italy, between November 2024 and February 2025. The diagnosis of IBS-D was established according to the Rome IV criteria. To reflect routine clinical practice, patient selection and management followed standard care protocols. All participants had undergone relevant clinical investigations within three months prior to enrollment, including complete blood count, blood chemistry (including anti-transglutaminase antibodies, total immunoglobulins, and thyroid function), stool analysis, colonoscopy and/or barium enema, all of which revealed no abnormal findings.

Exclusion criteria were minimal to ensure generalizability and included (1) presence of other organic GI diseases (inflammatory bowel disease, celiac disease, GI infections, GI cancers, or lactose intolerance); (2) significant systemic diseases (hepatic, renal, or cardiac dysfunction, diabetes mellitus, or cancer); (3) long-term use of antipsychotics or systemic corticosteroids; (4) use of antibiotics, probiotics, laxatives, or other medications affecting bowel movements within four weeks prior to this study; (5) recent colonoscopy, barium enema, or a history of acute gastroenteritis within two weeks prior to this study; (6) pregnancy or lactation; (7) drug or alcohol abuse during the study period; and (8) confirmed food allergies.

### 2.5. Clostridium butyricum CBM588 Probiotic Treatment

Patients were treated with a single-strain probiotic product containing *Clostridium butyricum* CBM588 (Butirrisan^®^, registered with Italian Health Ministry, dated 5 July 2022, registration code: 152311, Pharmextracta S.p.A., Pontenure, PC, Italy). The treatment regimen consisted of three tablets per day, taken during or shortly after breakfast, for a total of 8 weeks. Each tablet was reported to contain no less than 4.5 × 10⁵ CFU of *C. butyricum* at the expiry date (4 years from the manufacturing date). Microbiological analysis of the tablet content of the product batch used in our trial (performed at Teracell s.r.l., Cremona, Italy) revealed a CFU content of 1.5 × 10^7^ CFU of *C. butyricum*. The difference between what was reported on the label and the actual strain content was likely due to the overdosage, performed by the Japanese manufacturer, to guarantee the dose indeed declared at the expire date. Due to the lactose content (172.2 mg/tab), only lactose-tolerant individuals were enrolled in this prospective study. Lactose intolerance was established as (i) clinical intolerance (if reported by the patients) or (ii) by lactose via a breath test (if the patient was unable to report this aspect with certainty). Additionally, all participants were instructed to follow a low-fiber, low-residue diet throughout the study period.

### 2.6. Sample Size Calculation and Comparative Analysis with a Differently Treated Cohort

The sample size calculation was based on the total IBS-SSS score [14]. To detect a decrease of 25% with 95% power and a type I error of 0.05, our analysis required 200 participants (Appendix A). We therefore enrolled 205 subjects to take into consideration some possible drop-outs. For a better clinical interpretation of the results, we decided to compare the trend in the main outcomes with those obtained in a cohort of subjects (N = 200) overlapping in number, diagnosis, clinical features, and duration of observation attending our clinics the trimester before the start of our prospective study. As routine for our clinics, these 200 subjects were treated with trimebutine maleate; were suggested to follow a lactose-free, low-fiber, and low-residue diet; and were supplemented with probiotic products commonly available on the Italian market (Dicoflor^®^ (AG Pharma S.r.l., Rome, Italy) or Enterolactis plus^®^ (SOFAR S.p.A. Trezzano Rosa, Milan, Italy) or VSL3^®^ (VSL Pharmaceuticals Inc., Gaithersburg, MD, USA)) [20,21,22,23]. The retrospective analysis of the results of this cohort of subjects was performed respecting the Declaration of Helsinki [24] and the anonymity of the patients after obtaining their signed informed consent and having obtained the approval from the ethics committee for the retrospective use of these data.

### 2.7. Gut Microbiota Analysis

DNA was extracted from each sample using a QIAmp DNA stool kit following the manufacturer’s instructions (Qiagen Ltd., Strasse, Germany). Extracted DNA samples were kept at −20 °C until they were used for 16S rRNA analyses. Partial 16S rRNA gene sequences were amplified from the extracted DNA using the primer pair Probio_Uni (5′-CCTACGGGRSGCAGCAG-3′)/Probio_Rev (5′-ATTACCGCGGCTGCT-3′), which targets the V3 region of the 16S rRNA gene sequence. Illumina adapter overhang nucleotide sequences were then added to the partial 16S rRNA gene-specific amplicons, which in turn were further processed by employing the 16S Metagenomic Sequencing Library Preparation Protocol (Part #15044223 Rev. B; Illumina). Amplifications were carried out using a VeritiTM Thermocycler (Applied Biosystems; Foster City, CA, USA). The integrity of the PCR amplicons was analyzed by electrophoresis on a 2200 TapeStation Instrument (Agilent Technologies; Santa Clara, CA, USA). The PCR products obtained following the amplification of a section of the 16S rRNA gene were purified by a magnetic purification step involving Agencourt AMPure XP DNA purification beads (Beckman Coulter Genomics GmbH, Krefeld, Germany) in order to remove primer dimers. The DNA concentration of the amplified sequence library was estimated employing a fluorometric Qubit quantification system (Life Technologies; Carlsbad, CA, USA). Amplicons were diluted to 4 nM, and 5 µL of each diluted DNA amplicon sample was mixed to prepare the final pooled library. Paired-end sequencing (250 bp × 2) was performed using an Illumina MiSeq sequencer with MiSeq Reagent Kit v3 chemi-cals-600 cycles (Illumina Inc., San Diego, CA, USA). The FASTQ files were processed using QIIME2, as previously described [25,26]. Paired-end reads were merged, and quality control allowed the retention of sequences with a length between 140 and 400 bp, a mean sequence quality score > 25, and the truncation of a sequence at the first base if a low-quality sequence within a rolling 10 bp window was found. Sequences with mismatched forward and/or reverse primers were omitted. We defined 16S rRNA ASVs (amplicon sequence variants) at 100% sequence homology using DADA2 (version 1.26.0.) [27], and those with less than 10 sequences were filtered. The biological observation matrix (BIOM) obtained was analyzed using the summarize_taxa.py script in order to obtain the relative abundance of each taxonomic group for all samples. All reads were classified to the lowest possible taxonomic rank using QIIME2 and a reference dataset from the SILVA database v. 132 [28]. The microbial richness of the samples (α-diversity) was evaluated through the alpha_rarefaction.py script included in the QIIME2 (version 2024.10) software suite using the default parameters. Following sequencing, the FASTQ files were processed using a custom script for the QIIME2 software suite. Quality control led to retaining sequences with lengths between 100 and 400 bp and mean sequence quality scores of >20, while sequences with homopolymers >7 bp in length and mismatched primers were removed.

### 2.8. Statistical Analysis

The Wilcoxon Signed-Rank test was adopted to analyzed the “after versus before” results of the individual primary clinical outcomes of the prospective group (BSS, stool frequency, episodes of diarrhea, pain, bloating, bowel movements dissatisfaction, interference with QoL, IBS-SSS, and QoL). To reduce the risk of type I errors resulting from multiple comparisons and to evaluate the possible correlation between the clinical outcomes, principal component analysis (PCA) was used. PCA transforms a set of correlated variables into a new set of uncorrelated variables (principal components), while maintaining the maximum variance in the data. Particularly, this process can be beneficial for (i) dimensionality reduction (by reducing the variables to a few relevant principal components, the number of tests to be performed can be reduced, thus reducing the probability of incurring type I errors); (ii) removal of collinearity (PCA can mitigate the problem of multicollinearity between variables, improving the robustness of statistical models) [29,30]. To compare the biodiversity and taxonomy revealed by the gut microbiota analysis, we used, respectively, a paired *t*-test and the Wilcoxon–Mann–Whitney test. To evaluate the relationships between the clinical outcomes and the α-biodiversity and the shift in the microbial taxa, Partial Least Squares Regression (PLSR) was adopted [31]. Last, to compare the clinical outcomes between the prospective and the retrospective group, the Wilcoxon–Mann–Whitney test (also called Mann–Whitney U test) was used. JMP 14 (JMP14 is manufactured by JMP Statistical Discovery LLC, at SAS Institute (Cary, NC 27513, USA)) statistical software for macOS was used, and statistical significance was set at 95%.

## 3. Results

### 3.1. Demographic and Clinical Features at Enrollment

A total of 205 patients diagnosed with IBS-D were enrolled in this study. The mean age, weight, and duration since IBS diagnosis were 46.39 ± 16.76 years, 68.55 ± 13.32 kg, and 41.5 ± 67.37 months, respectively. Additional descriptive statistics, including minimum and maximum values and confidence intervals for the means, are provided in Appendix A.

Among the enrolled patients, 91 had comorbidities (Appendix A), with the most common being hypothyroidism (four patients), anxiety (four patients), dyslipidemia (three patients), polycystic ovary syndrome (three patients), hypertension (three patients), and thyroid disease (three patients). Additionally, 73 patients reported ongoing, or in some cases previously interrupted, therapies (Appendix A), the most frequent being thyroid hormone replacement (six patients) and combined estrogen–progestin therapy (three patients).

No enrolled patient reported nocturnal diarrhea episodes, unexplained weight loss, or the presence of mucus or blood in the stool within the past six months. All enrolled patients ended this study without discontinuing the proposed therapy.

### 3.2. Effect of C. butyricum CBM588 Probiotic Treatment on Primary Clinical IBS-D Outcome

As shown in Table 1, all primary outcomes—including the BSS, stool frequency, episodes of diarrhea, pain perception, bloating, bowel habit dissatisfaction, interference with QoL, IBS-SSS, and overall QoL—showed, in patients treated with a low-fiber and low-residue diet, significant improvement following the probiotic *C. butyricum* CBM588 treatment.

### 3.3. Treatment Effect on IBS Principal Components

To account for the potential type I errors associated with multiple comparisons and to better capture the correlations between the clinical parameters, principal component analysis (PCA) was applied. This statistical approach allows for dimensionality reduction, enabling a more comprehensive assessment of treatment effects while preserving the overall variability in the data. Additionally, PCA facilitates the visualization of the complex relationships between outcome measures, improving interpretability.

As shown in Figure 1, the distribution of key clinical outcomes (IBS-SSS, pain, bloating, interference with QoL, and overall QoL) at baseline (T0) and after 8 weeks of treatment (T8) along principal component 2 (PC2) significantly shifted between the two time points. This shift indicated that PC2 effectively captured meaningful clinical variations following treatment with *C. butyricum* CBM588, reinforcing its impact on symptom improvement and quality of life in patients with IBS-D.

### 3.4. Effect on Tolerability and Adherence

Treatment tolerability was rated highly, with a mean score of 8.95 ± 0.88 on a 0–10 scale. No severe or treatment-related adverse events were recorded. Mild adverse events were reported in six patients: three experienced bloating during the first week, and three reported borborygmi within the first five days.

Adherence was excellent, with an average compliance rate exceeding 95%. Only five patients missed doses due to forgetfulness—two missed two doses, and three missed one dose.

### 3.5. Effect of C. butyricum CBM588 on Gut Microbiota Structure

The changes in the α-diversity and gut bacterial composition at both the *phylum* and genus levels were analyzed at baseline (T0) and after 8 weeks of treatment (T8). As shown in Table 2 and Appendix A, treatment with *C. butyricum* CBM588, in patients treated with a low-fiber and low-residue diet, led to a significant increase in α-diversity (*p* = 0.008) and a global shift in the gut microbiota composition when measured at the phylum level (*p* = 0.022), without demonstrating a significant shift at the genus level (Appendix A).

Further analysis of the gut bacterial taxa at the two time points suggested a non-significant trend toward an increase in key acetate- and butyrate-producing genera, including *Bifidobacterium, Agathobacter, Butyricicoccus, Eubacterium, Subdoligranulum,* and *Coprococcus* (Appendix A).

### 3.6. Correlation Between the Primary Clinical Outcomes and k α-Biodiversity

Given the significant modification in the α-diversity following treatment, its correlation with the key clinical outcomes was assessed using Partial Least Squares Regression (PLSR). This statistical approach effectively manages multiple correlated variables, allowing for a clearer understanding of their individual contributions while reducing data complexity. By synthesizing all variables (clinical outcomes and α-diversity) into principal components, PLSR identified the most relevant variance and determines whether clinical improvements—such as reductions in diarrhea, pain, or bloating—were associated with an increased α-diversity.

The analysis confirmed a significant correlation between α-diversity and most of the clinical outcomes, particularly episodes of diarrhea and abdominal pain severity (Figure 2). Additional associations were observed with abdominal pain frequency, bloating, and bowel habit dissatisfaction (Appendix A).

Among these variables, diarrhea episodes, abdominal pain severity, and bowel dissatisfaction exhibited a Variable Importance for Projection (VIP) score > 1, identifying them as key drivers of biodiversity variation. Notably, diarrhea episodes per day had the greatest impact, reinforcing the idea that improvements in this clinical outcome are directly linked to increased gut microbial richness. While abdominal pain frequency and bloating had a smaller influence in the model, their contribution remained relevant.

As shown in Figure 2, a significant correlation was observed between the gut microbiota richness (ASV) and key clinical parameters. The left graph illustrates the relationship between episodes of diarrhea and microbial richness, where a reduction in diarrhea episodes is associated with an increase in richness. The right graph presents the correlation between abdominal pain severity and microbial richness, indicating that lower pain severity corresponds to higher richness.

The negative coefficient values in both graphs suggest that as diarrhea episodes and pain severity decreased, microbial richness increased. Additionally, VIP scores greater than one confirmed that these clinical parameters were key contributors to the gut microbiota variability in the Partial Least Squares Regression (PLSR) model.

### 3.7. Correlation Between Primary Clinical Outcomes and Butyrate-Producing Taxa

To assess the potential correlations between the primary clinical outcomes and specific butyrate-producing bacterial taxa, the Partial Least Squares Regression (PLSR) method was applied. The analysis focused on *Agathobacter*, *Butyricicoccus*, *Coprococcus*, and *Eubacterium*, which increased following treatment with *C. butyricum* CBM588.

As illustrated in Figure 3, the regression trend indicates a correlation between the combined butyrate-producing *taxa* (X-axis) and the primary clinical outcomes (Y-axis). This suggests that the changes in the gut microbiota composition, particularly an increase in butyrate-producing bacteria, were associated with improvements in key clinical parameters.

Figure 3 shows that an increase in the butyrate-producing bacteria (*Agathobacter*, *Butyricicoccus*, *Coprococcus*, and *Eubacterium*, X-axis) was associated with an improvement in the primary clinical outcomes (Y-axis), including reductions in diarrhea episodes, abdominal pain severity and frequency, bloating, and bowel dissatisfaction. The positive trend in the regression line suggests that a higher abundance of these butyrate-producing *taxa* correlated with better clinical outcomes.

However, the four butyrate-producing taxa did not contribute equally to the observed improvements. Based on the Variable Importance for Projection (VIP) values, only three taxa—*Agathobacter*, *Butyricicoccus*, and *Coprococcus*—showed significant associations, though their individual effects remained modest (Appendix A).

Further analysis examined whether an increase in a single taxon consistently predicted symptom improvement. As shown in Figure 4, *Agathobacter* had no significant effect on diarrhea episodes or bowel habit dissatisfaction but was positively associated with reductions in abdominal pain severity and frequency while negatively impacting bloating. *Butyricicoccus* did not significantly influence diarrhea, pain severity, bloating, or bowel habit dissatisfaction; however, it showed a positive association with reduced pain frequency. Similarly, *Coprococcus* had no significant effect on diarrhea but positively correlated with improvements in pain severity, pain frequency, and bowel habit dissatisfaction, while being negatively associated with bloating. In contrast, Eubacterium did not exhibit any significant correlation with the clinical parameters analyzed (Appendix A).

These findings suggest that while some butyrate-producing bacteria contribute to symptom improvement, their effects vary across different clinical parameters, underscoring the complexity of the gut microbiota interactions in IBS-D.

### 3.8. Comparison of the C. butyricum CBM588 Probiotic Treatment with a Retrospective IBS-D Control Group

For a more comprehensive clinical interpretation of the findings, the primary clinical outcomes presented in Table 1 were compared with those obtained retrospectively from a control cohort of 200 patients with IBS-D with comparable clinical characteristics. This retrospective cohort was treated with trimebutine maleate; advised to follow a lactose-free, low-fiber, and low-residue diet; and supplemented with various commercially available probiotic products, as detailed in the Section 2.

The mean age, weight, and duration since IBS diagnosis in the retrospective cohort were 46.81 ± 6.51 years, 72.32 ± 11.20 kg, and 38.4 ± 6.20 months, respectively, with no significant differences compared to those of the prospective cohort.

Similarly, the clinical outcomes were highly comparable between the groups (Table 3) and showed no statistically significant differences (Appendix A).

## 4. Discussion

This prospective, pragmatic, single-arm, open-label, interventional study involving 205 patients with IBS-D demonstrated that probiotic therapy with the *C. butyricum* CBM588 strain, described both as an important butyrate producer and capable of determining significant clinical impact [32], can lead to significant clinical improvements in patients treated with a low-fiber and low-residue diet. These findings align with previous observations in Chinese patients [14], suggesting that similar effects can be achieved in Caucasian populations.

### 4.1. Clinical Efficacy of C. butyricum CBM588 in Treating IBS-D

The probiotic intervention resulted in a reduction in diarrhea episodes by more than 80%, a 60% decrease in evacuation frequency, and a 30% improvement in Bristol Stool Scale scores. Additionally, abdominal pain decreased by 60%; bloating by over 50%; and dissatisfaction with bowel habits and interference with daily life improved by approximately 50% and 60%, respectively. Overall, IBS severity and quality of life improved by more than 50%. These findings were further validated through principal component analysis (PCA), confirming the robustness of the observed effects.

### 4.2. Comparison with Retrospective IBS-D Control Group

Although this pragmatic study did not include a prospective control group, a retrospective evaluation was conducted on 200 patients with IBS-D treated at the same hospital in the previous quarter. These patients received trimebutine (150 mg every 8 or 12 h); advised to follow a lactose-free, low-fiber, and low-residue diet, as also suggested in the Rome III and IV criteria [33]; and supplemented with commercially available probiotics. The clinical outcomes between the prospective and retrospective cohorts were highly comparable, suggesting that the observed improvements in the CBM588 group were consistent with the expected clinical response in IBS-D.

While the primary difference between the groups was the probiotic intervention vs. trimebutine treatment, another noteworthy distinction was lactose exposure. The prospective cohort was limited to non-lactose-intolerant individuals, whereas the retrospective group followed a lactose-free diet. However, it remains unclear whether this minor dietary variation influenced the clinical outcomes, and it is unlikely to have substantially altered this study’s findings.

### 4.3. Impact on Gut Microbiota

A subset of 19 patients (approximately 10%) underwent gut microbiota analysis before and after CBM588 therapy. Although initially planned for 20 participants, one patient was excluded due to missing baseline stool samples. Despite the small sample size, the analysis revealed a significant increase in α-diversity and a trend toward an increase in certain butyrate-producing taxa.

Further statistical validation using the PLSR method confirmed a significant correlation between the clinical improvements and microbiota changes, particularly with α-diversity and the taxa *Agathobacter*, *Butyricicoccus*, and *Coprococcus*. These findings align with those of prior research in IBD, where diarrheal symptoms have typically prevailed [34,35,36,37,38,39], and IBS-D, where enhanced biodiversity and butyrate production have been associated with symptom relief [40,41].

Interestingly, no detectable increase in *C. butyricum* itself was observed. Since the analysis relied on 16S rRNA sequencing, which classifies bacteria at the genus level, *C. butyricum* was grouped under *Clostridium sensu stricto* 1 (Bacillota; former Firmicutes). The absence of a measurable increase does not necessarily indicate a lack of colonization, as the method does not distinguish between *C. butyricum* and other *Clostridium* species, such as *C. perfringens*. A metagenomic shotgun sequencing approach would be required to overcome this limitation. If *C. butyricum* replaced *C. perfringens*, the relative abundance of *Clostridium sensu stricto* 1 would remain unchanged, which aligns with the observed findings.

### 4.4. Reduction in Abdominal Pain and Potential NLRP6 Modulation

A particularly noteworthy finding was the significant reduction in abdominal pain in the CBM588-treated cohort. This effect was not as pronounced as in the Chinese study [14] and may be linked to the probiotic’s influence on NLRP6 [42], an intracellular receptor involved in immune regulation and inflammasome activation [43]. A previous study suggests that NLRP6 modulates the caspase-1 and NFκB pathways, both of which play key roles in inflammatory responses in IBS [44]. Although speculative, these mechanisms warrant further investigation.

### 4.5. Association Between Butyrate Producers and Clinical Outcomes

This study confirmed a significant correlation between increased butyrate-producing bacteria and symptom improvement, except for bloating, which appeared to worsen. One possible explanation is the fermentative activity of butyrate producers, which leads to increased gas production, contributing to bloating. This was previously observed in high-FODMAP diets, where excess fermentation by certain gut bacteria exacerbated bloating symptoms [45]. From a dietary perspective, patients were advised to reduce high-fiber foods, but a structured low-FODMAP diet was not implemented [22]. Recent research suggested that a low-FODMAP diet, particularly in individuals with a Bacillota-dominant microbiota, improved IBS symptoms by reducing fermentation-associated bloating [46]. However, without detailed dietary records, the extent to which dietary factors influenced this study’s outcomes remains unknown. Interestingly, the literature suggests that the bacterial pathway of the fermentation of oligosaccharide fiber and polyols is particularly present in the Bacillota cluster [47], and a low-FODMAP diet tends to decrease Bacillota and butyrate producers [48], yet this study observed an increase in butyrate-producing taxa. This suggests that probiotic supplementation had a more significant impact than dietary modification alone.

### 4.6. Limitation of This Study

Despite its strengths, this study has several limitations. The lack of a prospective control group is a key limitation; while a retrospective comparison was made, a randomized controlled trial (RCT) with a placebo group would provide more robust evidence. Although the retrospective cohort consisted of patients treated in the previous quarter with a different pharmacological approach, its reliability was inherently limited due to differences in study conditions and potential uncontrolled variables. The choice to not include a prospective randomized control group was primarily driven by the pragmatic nature of this study, which aimed to evaluate intervention outcomes under routine clinical conditions, as typically experienced in real-life medical practice. Constraints on patient recruitment capacity and ethical feasibility for placebo withholding in symptomatic patients also influenced this decision. The open-label design is another notable limitation, as knowing the treatment being administered could have influenced symptom reporting, introducing an element of expectation bias. However, a blinded, placebo-controlled format was not logistically feasible in the real-world outpatient setting in which this study was conducted. Open-label designs are commonly employed in pragmatic research to maintain operational viability while reflecting everyday clinical decision making. Nevertheless, we acknowledge the potential for bias, which supports the need for future blinded studies.

The short duration of this study is an additional limitation, as the 8-week intervention did not include a wash-out period or long-term follow-up. Consequently, it remains unknown whether the observed effects would have been sustained beyond 8 weeks, how long the benefits persisted after treatment, or if an earlier clinical response (e.g., after 4 weeks) would have been detected. The impact of probiotic dosage is another uncertainty, as the three-tablet-per-day regimen was adopted based on manufacturer recommendations (Miyarisan Pharmaceutical Co., Ltd., Tokyo, Japan). Whether higher or lower doses (e.g., six or nine tablets per day) would have yielded different clinical outcomes remains unexplored.

Dietary intake, although partially controlled, was not strictly monitored, making it difficult to determine the precise influence of dietary modifications on the gut microbiota composition and symptoms. While patients were advised to reduce their intake of high-fiber foods, a structured low-FODMAP diet was not implemented, and no food diary was maintained. This made it challenging to assess whether nutritional factors contributed to the observed effects or if the probiotic intervention played a predominant role. Given the well-established role of diet in modulating the gut microbiota and gastrointestinal symptoms, the absence of structured dietary assessments—such as 24 h recall, food frequency questionnaires, or validated dietary tracking tools—is a potential confounding factor. Future trials should incorporate formal dietary monitoring to isolate the intervention effects more clearly.

Furthermore, microbiota analysis was performed on only 19 patients, which was a small sample size and limited the ability to draw definitive conclusions about microbial shifts. This limitation was primarily due to the exploratory nature of this study, conducted in real-life clinical practice settings where extensive sampling was not always feasible. Resource constraints and the need to minimize participant burden also influenced the decision to analyze the microbiota in a representative subset rather than the full cohort. Additionally, the 16S rRNA sequencing approach restricted bacterial classification to the genus level, preventing the species-level identification of *C. butyricum*. More precise metagenomic techniques, such as shotgun sequencing, would have been necessary to confirm whether *C. butyricum* colonization occurred or if it modulated gut microbiota composition indirectly.

Despite these limitations, this study provides valuable insights into the potential role of butyrate-producing probiotics in IBS-D management. The observed clinical improvements in the CBM588 group were comparable to those in patients treated with trimebutine, a standard pharmacological therapy for IBS-D. This finding supports the therapeutic relevance of butyrate production, which distinguishes CBM588 from other commonly studied probiotics such as *Lactobacillus* and *Bifidobacterium* strains, which lack butyrate-producing capabilities.

Previous meta-analyses of probiotic therapy in IBS have reported conflicting results, likely due to the heterogeneity of the probiotic strains studied. Most of these trials focused on non-butyrate-producing genera, including *Lactobacillus*, *Bifidobacterium*, *Lactococcus*, *Enterococcus*, *Streptococcus*, and *Leuconostoc*, which exert different mechanistic effects on gut health [49,50,51,52,53]. Importantly, *C. butyricum* utilizes a distinct metabolic pathway for butyrate production, either via the acetate dimerization pathway (butyryl-coenzyme A: acetate-coenzyme A transferase, as in *Faecalibacterium* species) or through ex novo butyrate synthesis (butyrate kinase, as in *C. butyricum*) [54]. Given these differences, it is possible that butyrate-producing probiotics provide a distinct advantage in IBS-D management, potentially altering the current probiotic paradigm.

Although prospective, open-label, and interventional, our study is however preliminary, and it lays the groundwork for future randomized, double-blind, placebo-controlled trials to confirm these findings. A well-structured RCT would help determine the optimal probiotic dosage, duration of treatment, and long-term sustainability of effects, providing stronger evidence for the clinical utility of CBM588 in IBS-D treatment.

## 5. Conclusions

Despite the limitations of a pragmatic study design and the small number of fecal samples analyzed, the findings suggest that *Clostridium butyricum* CBM588, in patients treated with a low-fiber and low-residue diet, has a significant impact on the primary clinical symptoms of IBS-D. The probiotic treatment was associated with improvements in diarrhea episodes, abdominal pain, bloating, and bowel dissatisfaction, while also inducing modifications in the gut microbiota composition. Specifically, increases in the α-diversity and the relative abundance of butyrate-producing taxa were observed.

Moreover, the changes in the gut microbiota composition correlated significantly with symptom improvement, with the exception of bloating, which appeared to increase in parallel with the rise in *Agathobacter*, *Butyricicoccus*, and *Coprococcus*. These findings highlight both the potential therapeutic benefits and complexities associated with modulating the gut microbiota in IBS-D, emphasizing the need for further controlled trials to confirm these effects and optimize probiotic interventions for clinical use.

## Figures and Tables

**Figure 1 microorganisms-13-01139-f001:**
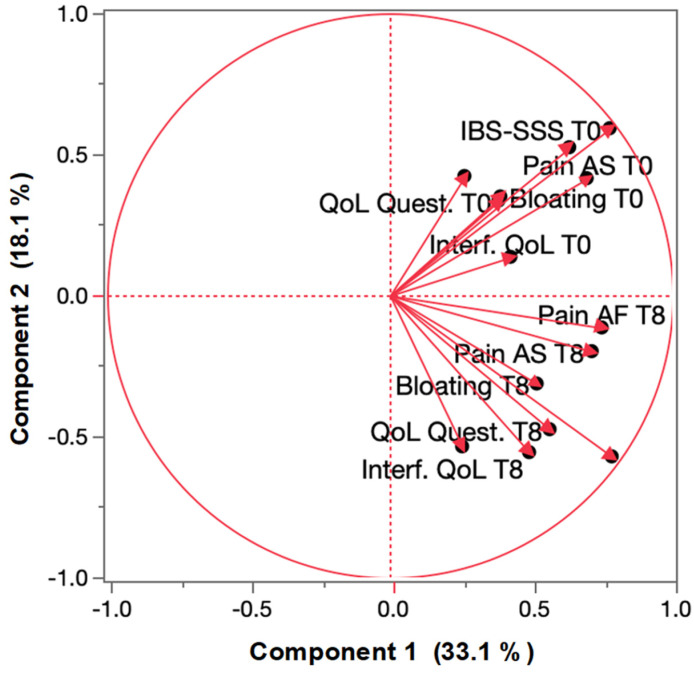
*Clostridium butyricum* CBM588 probiotic treatment on principal components of IBS-D in primary study outcome measures. T0: baseline; T8: after 8 weeks of treatment; QoL: quality of life; IBS-SSS: Irritable Bowel Syndrome—Symptom Severity Scale; Quest.: questionnaire; Interf.: interference.

**Figure 2 microorganisms-13-01139-f002:**
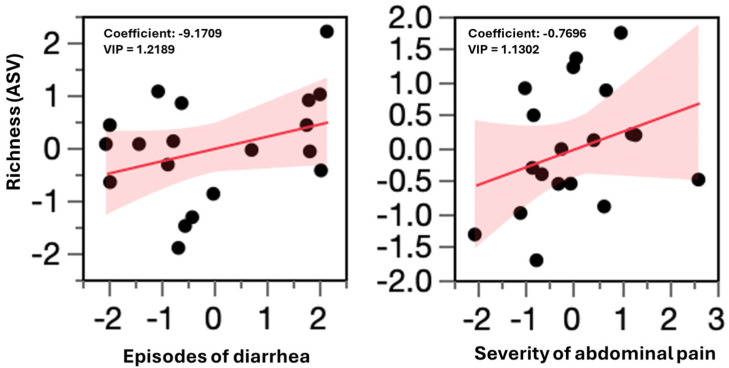
Correlation between gut microbiota richness and episodes of diarrhea and severe abdominal pain.

**Figure 3 microorganisms-13-01139-f003:**
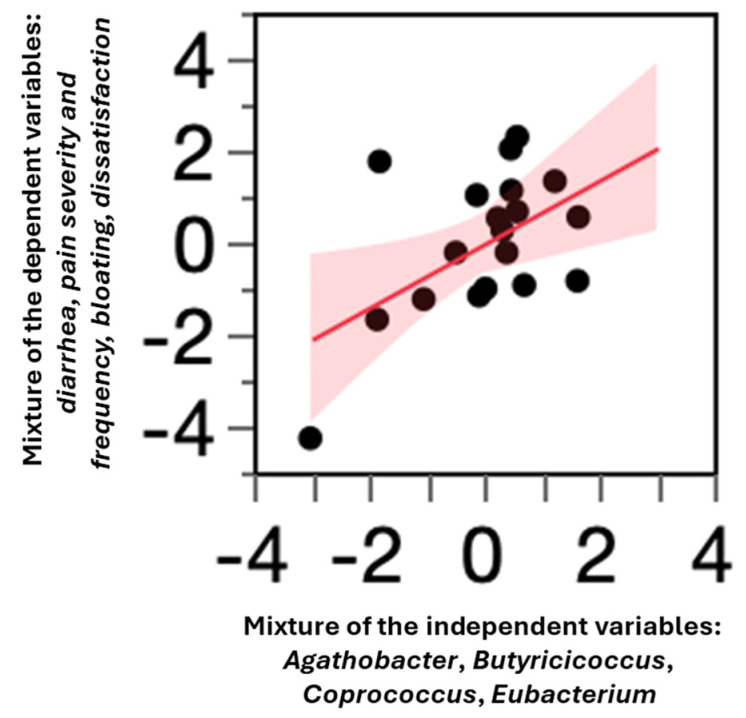
Correlation between butyrate-producing taxa and primary clinical outcomes.

**Figure 4 microorganisms-13-01139-f004:**
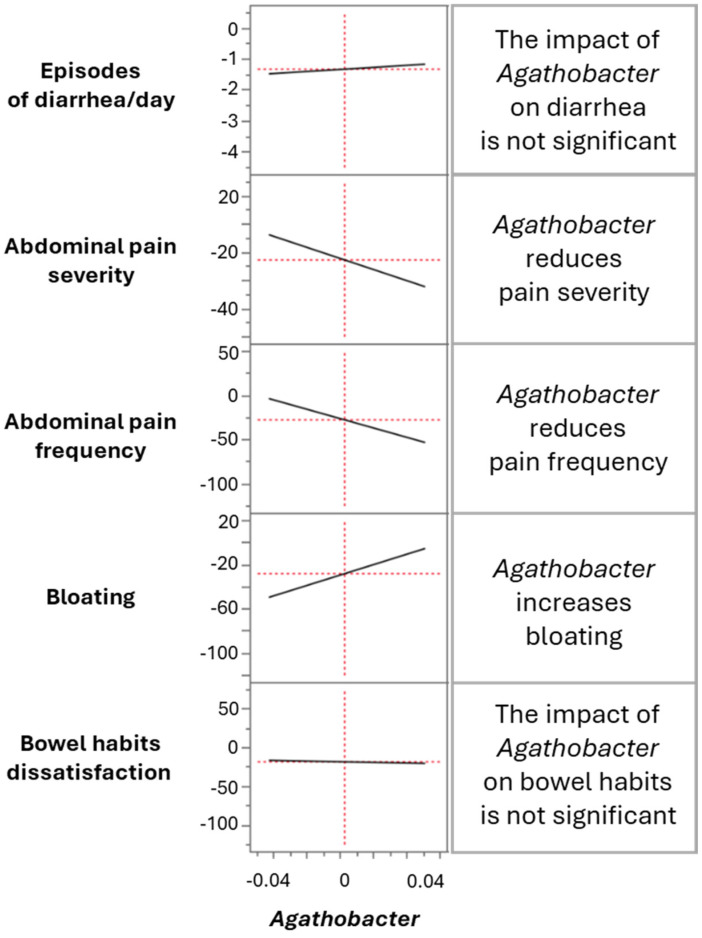
Differential impact of *Agathobacter* on primary clinical outcomes. The X-axis represents the relative abundance of *Agathobacter*, while the Y-axis displays five different primary clinical outcomes, each scaled according to its respective measurement.

**Table 1 microorganisms-13-01139-t001:** Effect of 8-week *Clostridium butyricum* CBM588 probiotic treatment on the primary study outcomes measured in patients with IBS-D.

Clinical Variable	M ± SD	Variation (%)	*p*
Bristol Stool Score T = 0	5.93 ± 1.07		
Bristol Stool Score T = 8 w	4.19 ± 0.95	−29.4	<0.001
Stool frequency per day T = 0	2.93 ± 1.29		
Stool frequency per day T = 8 w	1.27 ± 0.63	−56.7	<0.001
Episodes of diarrhea per day T = 0	2.09 ± 1.50		
Episodes of diarrhea per day T = 8 w	0.32 ± 0.68	−84.7	<0.0001
Pain (severity) T = 0	43.37 ± 27.74		
Pain (severity) T = 8 w	17.95 ± 14.68	−58.6	0.001
Pain (frequency; out of 10 days) T = 0	38.59 ± 28.59		
Pain (frequency; out of 10 days) T = 8 w	13.32 ± 11.70	−64.5	<0.0001
Bloating T = 0	65.85 ± 27.28		
Bloating T = 8 w	30.73 ± 19.35	−53.4	<0.0001
Bowel habits dissatisfaction T = 0	75.80 ± 18.39		
Bowel habits dissatisfaction T = 8 w	40.20 ± 22.84	−47.9	<0.0001
Interference with QoL T = 0	59.12 ± 19.81		
Interference with QoL T = 8 w	23.32 ± 16.80	−60.6	<0.0001
IBS-SSS T = 0	282.39 ± 79.05		
IBS-SSS T = 8 w	125.90 ± 55.83	−55.5	<0.0001
QoL T = 0	32.91 ± 8.04		
QoL T = 8 w	15.87 ± 4.31	−51.8	<0.0001

M ± SD: mean ± standard deviation; T = 0: baseline; T = 8 w: after 8 weeks of treatment; QoL: quality of life; IBS-SSS: Irritable Bowel Syndrome—Symptom Severity Scale.

**Table 2 microorganisms-13-01139-t002:** Shift in α-biodiversity after 8 weeks of *C. butyricum* CBM588 probiotic treatment.

	M ± SD	Variation (%)	*p*
α-biodiversity T = 0	118.74 ± 49.23	-	-
α-biodiversity T = 8 w	143.37 ± 62.82	+20.7	0.008

M ± SD: mean ± standard deviation.

**Table 3 microorganisms-13-01139-t003:** Clinical outcomes in the retrospective control IBS-D cohort treated with the usual care.

Clinical Variable	M ± SD	Variation (%)	*p*
Bristol Stool Score T = 0	6.31 ± 0.95		
Bristol Stool Score T = 8 w	3.98 ± 1.50	−36.9	<0.001
Stool frequency per day T = 0	3.22 ± 0.65		
Stool frequency per day T = 8 w	1.77 ± 0.22	−45.1	<0.001
Episodes of diarrhea per day T = 0	2.42 ± 0.35		
Episodes of diarrhea per day T = 8 w	0.95 ± 0.13	−60.8	<0.0001
Pain (severity) T = 0	47.09 ± 16.80		
Pain (severity) T = 8 w	22.15 ± 13.05	−58.6	<0.01
Pain (frequency; out of 10 days) T = 0	43.24 ± 6.27		
Pain (frequency; out of 10 days) T = 8 w	21.18 ± 6.30	−51.0	<0.0001
Bloating T = 0	60.37 ± 19.43		
Bloating T = 8 w	32.12 ± 14.31	−46.8	<0.0001
Bowel habits dissatisfaction T = 0	66.39 ± 19.79		
Bowel habits dissatisfaction T = 8 w	35.63 ± 15.17	−46.4	<0.0001
Interference with QoL T = 0	65.55 ± 19.87		
Interference with QoL T = 8 w	31.53 ± 14.42	−51.9	<0.0001
IBS-SSS T = 0	281.73 ± 38.35		
IBS-SSS T = 8 w	141.28 ± 27.53	−49.9	<0.0001
QoL T = 0	34.33 ± 5.12		
QoL T = 8 w	15.10 ± 2.62	−56.1	<0.0001

M ± SD: mean ± standard deviation; T = 0: baseline; T = 8 w: after 8 weeks of treatment; QoL: quality of life; IBS-SSS: Irritable Bowel Syndrome—Symptom Severity Scale.

## Data Availability

The original contributions presented in this study are included in this article/Appendix A. Further inquiries can be directed to the corresponding authors.

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
