# Peer review of "Irritable Bowel Syndrome with Diarrhea (IBS-D): Effects of Clostridium butyricum CBM588 Probiotic on Gastrointestinal Symptoms, Quality of Life, and Gut Microbiota in a Prospective Real-Life Interventional Study"

_microorganisms, 2025, doi:10.3390/microorganisms13051139_

Round 1

Reviewer 1 Report

Comments and Suggestions for Authors

A presented pragmatic clinical trial -  is a clinical trial that focuses on correlation between treatments of Irritable Bowel Syndrome with Diarrhea by  Clostridium butyricum CBM588 Probiotic and outcomes in real-world health system practice rather than focusing on proving causative explan.  

The authors have demonstrated the clinical efficacy of this probiotic in a fairly large number of patients. There were a decrease in diarrhea and the frequency and quality of bowel movements according to the Bristol Stool Scale and an increase of life quality . Examinations and patient questionnaires were also used for this purpose.  The placebo group was not used. Patients receiving therapy with symptomatic agents, such as trimebutine, were selected as the control group

Probiotic therapy for 8 weeks proved to be clinically more effective.  This is the main effect of the authors and an important one. But the comparison was not quite correct, and it is unclear which groups were compared in the end.

Using the butyrate-producing probiotic strain Clostridium butyricum CBM588, the researchers note an increase in other butyrate producers who find themselves in more favorable conditions with additional supply of their key substrate to the cooling medium. This is an interesting phenomenon that may find serious prospects in research and practical application in the future.

At the same time there are number of significant shortcomings and questionable points in the work.

  1. An excessively small dose of probiotic was used,  4.5 × 10⁵ CFU of C. butyricum 152 in fact, the probiotic synbiotic culture and  prebiotic lactulose  were used, but the dose was less than 1 million bacteria, that is, less than the "required quorum".
  2.  Despite the fact that only 19 samples were examined before and after therapy  authors have a chance to better present the results of the microbiome study. 
  3.  It is strange that it was possible to detect changes only in the representation of butyrate producers and alfa-biodiversity. But we would like to see these  and may be other data in the form of graphs. 
  4. Interesting data on correlation analysis must be shown with an indication of the correlation coefficient.
  5. Firmicutes - the old name of the phylum is now correctly called "Bacillota"
  6. In fact, the work mainly proves the clinical effectiveness of the synbiotic used. The authors can try to make the effect of the synbiotic on the intestinal microbiota more visual, show the representation of other bacteria that are often prevalent in IBS, and compare it in the discussion section with the effect of other probiotics on the microbiota and clinical and laboratory data in the pathology under consideration.

Author Response

Thanks

Amjad

Reviewer 2 Report

Comments and Suggestions for Authors

This manuscript documents an investigation into the effects of the butyrate-producing probiotic Clostridium butyricum CBM588 in patients with diarrhea-predominant irritable bowel syndrome (IBS-D). The findings provide some insights into the potential beneficial role of targeted probiotics in gastrointestinal health, particularly in this case with a focus on their impact on microbiota diversity and the potential benefits of butyrate production by the probiotic.

 The use of CBM588 appears to have improved clinical symptoms, quality of life, and some key gut microbiota parameters over the 8-week intervention period of the study. The authors have introduced a comparison with a retrospective cohort receiving standard care (trimebutine and other commercially available probiotics) to provide a standard treatment  context to their intervention, and this indicates that the use of CBM588 has achieved comparable symptomatic relief. The increases in microbiota diversity and butyrate-producing bacteria however appear to distinguish the use of CBM588 from existing probiotic options and it is contended that this perhaps can confer additional mechanistic benefits beyond symptom control.

The authors have appropriately discussed some of the major limitations of their study that may have impacted upon the strength of the conclusions to be drawn. Perhaps some further explanation of the reason(s) for these omissions would be helpful? The lack of a prospective, randomized control group is clearly important, as is the small subset of patients analysed for microbiota changes. Also, the open-label design introduces some potential for bias. The present study nevertheless does provide a strong initiative for follow-up studies of longer duration and these should also include more detailed  monitoring of the potential for  any dietary  influences on the observed microbiota outcomes.

Author Response

Thanks

Amjad

Reviewer 3 Report

Comments and Suggestions for Authors

Dear Authors,

I have read carefully Your manuscript entitled “Irritable Bowel Syndrome with Diarrhea (IBS-D): Effects of Clostridium butyricum CBM588 Probiotic on GI Symptoms, Quality of Life, and Gut Microbiota in a Pragmatic Clinical Study”.

The Authors conducted a prospective, open label study comparing a group of patients prospectively enrolled and treated with a probiotic containing C. butyricum MIYAIRI 588 in addition to a low-fiber, a low-residue diet (administered to patients without lactose intolerance), to a cohort of patients retrospectively enrolled, who recieved a treatment based on trimebutine maleate, along with a lactose-free, low-fiber, and low-residue diet, and supplemented with various commercially available. The Authors reported no significant differences in outcomes were observed between the two groups at baseline (T0) or after 8 weeks of treatment (T8), essentially indicating that the two treatment arms demonstrated comparable efficacy.

At the end of the discussion, the Authors stated that the study was observational, while in the other section of the article they refer to ita as a prospective, open lable, interventional study. Upon reviewing the registration on ClinicalTrials.gov, the study is labled as observational. Please carefully review the entire manuscript and confirm whether the study was designed as a prospective observational study or a prospective, open-label, interventional study, as this distinction significantly affects both the manuscript structure and the interpretation of the study outcomes.

Furthermore, in the abstract, results, discussion and conclusion sections, it must be clearly stated that the improvement in symptoms needs to be attributed to the combination of probiotic and low-fiber, low-residue diet and not to the probiotic alone.

Introduction

Lines 53-54. A prevalence of 15-20% seems quite high for IBS. The cited article (ACG guidelines) reported a prevalence “of approximately 4.4%–4.8% in the United States, United Kingdom, and Canada and affects most commonly women and individuals younger than 50 years”. Therefore, I would suggest to update this prevalence data according to most recently published evidence. Furthermore, a recent Rome Foundation epidemiological study (DOI: 10.1053/j.gastro.2020.04.014) reported a worldwide prevalence of Rome IV IBS around 4%.

Lines 54-56. Regarding the definition of IBS, the Authors referred to Rome III criteria citing them and including the defecation-related abdominal discomfort in their definition. Later in the manuscript, however, they refer to Rome IV criteria, which do not take into account abdominal discomfort. The Rome IV criteria have been used also to diagnose IBS in enrolled patients. Please consider fixing this issue in order to give more consistency with the definition of IBS.

Line 59. Before “Rome IV criteria” there is an extra space.

Line 77. The C. butyricum MIYAIRI 588 is considered a “food”, Checking online, the complete name is “food supplement”. Please consider updating this definition in the introduction.

Materials and methods

There are some chapter sections in Italic and some other no, please unify the stile.

Lines 85-86. This was an open lable study and therefore the efficacy of the product can’t be adequately assessed. I suggest referring instead to "perceived efficacy”. Moreover, the comparison Even the comparison with previously treated patients does not fully eliminate the potential bias arising from a placebo effect.

Line 104., has the standardized tolerability scale been previously used in papers? Is it validated? Please specify in this section of the manuscript.

Lines 105-119. Please provide adequate bibliographyc reference for the scales and questionnaires.

Lines 136-137. The blood chemistry included celiac screening with transgltaminase antibodies plus total immuniglobulins and thyroid function? If so, please specify in the text. Why the Authors have decided to perform both barium enema and colonoscopy in all patients?

Line 151. Patients took all three capsules during breakfast or during also the other meals? Usually this probiotici s reccommended 1 to 3 tablets daily during the main meals. Please clarify this point.

Line 152-154. Therefore have Authors considered lactose intollerance as an exclusion criteria? How was lactose tolerancy established? Through a breath test? Please clarify this point.

Line 154-155. “In addition to the probiotic, patients were instructed to follow a low-fiber, low-residue diet throughout the study period”. In some IBS patients, the reduction of fermentable fibers can reduce the symptoms. Therefore, we cannot conclude that the observed effect was linked to the probiotic supplementation or the reduction in dietary fibers. Why Authors have decided to add low-fiber diet to the probiotic? Usually fibers are considere prebiotics and therefore, in adequate quantities, may help reaching a better microbiota composition. Maybe to make the two groups more comparable? Please clarify this point.

Line 210. Why have Authors decided to use a non-parametric test for their primary outcome rather than a paired t-test? Did Authors assessed the normality distribution of their data? Furthermore, to compare alfa-diversity of gut microbiota, Authors have used a paired t-test. Please update the statistycal analysis section. Finally, it is not described in this section how treated patients were compared with the control group. In this case, the test must have been different compared with the ones reported by Authors, considering that the two groups of patients are not paired.

Did Authors considered any rescue therapy for patients? If yes, what medications and at what maximum dosage were allowed?

Results

How many patients dropped out from the study or discontinued the medication?

Table 1. Authors used a non parametric test as the Wilcoxon Signed-Rank Test to test for significancy but then expressed the continuous variables as mean and SD rather than median and Interquartile range (more suitable for non normal data). Please consider modifying or the used test or the presented data.

Table 1. Before “Pain (frequency) T0 there is a 0.

Tables and figures

I think that, to allow a better understanding of the whole paper, comorbidities and therapies reported in supplementary material should be translated in English. Furthermore, in some cases, the Authors used the commercial name of the product and in some other cases the active ingredient. Please standardize reporting only the active ingredient. Moreover, some medications in the list are among the exclusion criteria (e.g. rifaximin, an antibiotic). Please specify if these medications were discontinued before the study as written in the methods section.

Tables both in the main body of the manuscript and in supplementary materials need consistency in how they are formatted (character, color, style of table). Please fix this issue.

Author Response

Thanks

Amjad

Round 2

Reviewer 3 Report

Comments and Suggestions for Authors

Thank you very much for answering all raised comments.